# Global, regional, and national burden of cataract: A comprehensive analysis and projections from 1990 to 2021

Lixia Lin[1,2], Yongshun Liang[1], Guiyang Jiang[3], Qingqiao Gan[1], Tianqi Yang[1], Peipei Liao[1], Hao Liang [1]*

1 Ophthalmology Department, The First Affiliated Hospital of Guangxi Medical University, Nanning, GuangXi, China, 2 Ophthalmology Department, The First People's Hospital of Nanning, Nanning, Guangxi, China, 3 Rehabilitation Medicine Department, The First Affiliated Hospital of Guangxi Medical University, Nanning, Guangxi, China

* liangh@stu.gxmu.edu.cn

## Abstract

### Objective

Cataract is the most prevalent cause of blindness. Surgery remains the only effective and widely accepted treatment; early diagnosis and intervention can significantly prevent blindness. Hence,Understanding the current epidemiological status of cataract is crucial for formulating better healthcare policies and effectively preventing blindness due to cataract.

### Design

This study utilizes Global burden of Disease (GBD) 2021 data to conduct an in-depth analysis of the burden of cataract from 1990 to 2021, including gender disparities, risk factors, and the relationship between Socio-Demographic index (SDI) and disease burden. Additionally, we performed a frontier analysis of Disability-Adjusted Life Years (DALYs) due to cataract from 1990 to 2021. Finally, we used the BAPC model to project the burden of cataract by gender from 2022 to 2030.

### Results

The study revealed that the global burden of cataract remains significant. Worldwide, the Estimate Annual Percentage Change (EAPC) for cataract prevalence was 0.2117([95% CI] 0.1172–0.3063); the EAPC for cataract DALYs is −0.4798([95%CI] −0.5766--0.3828). Predominantly affecting females, individuals aged 50 and older, and those in medium-low and low SDI regions. Furthermore, the Bayesian Age-Period-Cohort (BAPC) model forecast a gradual decline in the global burden of cataract over the next nine years.

**Data availability statement:** Data are available from the Global burden of disease 2021 database. The data underlying the results presented in the study are available from GBD2021 database(https://ghdx.healthdata.org/gbd-2021).

**Funding:** This work was funded by National Natural Science Foundation of China (Grant Nos. 81960174 and 81360146).

**Competing interests:** The authors have declared that no competing interests exist.

## Conclusion

This study utilized GBD 2021 to provide an in-depth analysis of the current global disease burden of cataracts. The results showed that although the Age-Standardized Rate (ASR) of DALYs decreased, the overall cataract Number still showed an increasing trend from 1990 to 2021 and 2022–2030.

## Introduction

Cataract is a condition characterized by lens opacification that impairs vision. Symptoms include blurred vision, reduced contrast sensitivity, and monocular diplopia. The condition is primarily managed through surgical intervention [1,2]. In 2020, approximately 94 million individuals aged 50 and older worldwide experienced blindness or visual impairment due to cataract. This number is expected to rise as the population ages, leading to an increase in the number of cataract patients aged 60 and above [2,3]. Cataract is primarily classified into congenital and acquired types. Among acquired cataracts, age-related cataract is the most common and a major cause of visual impairment and blindness. The pathogenesis of cataract remains not fully elucidated, although current understanding mainly focuses on aging and oxidative stress [3]. Additionally, risk factors for this condition include ocular diseases other than cataract, diabetes, hypertension, obesity, ultraviolet radiation, and smoking [4–7]. Currently, there are no effective pharmacological treatments for cataract; surgery remains the sole method for curing the condition. This is largely because the pathogenesis of cataract is multifactorial, involving oxidative stress, protein aggregation, and other complex mechanisms. Additionally, the delivery of drugs to the lens is challenging due to the blood-aqueous barrier, which limits the efficacy of pharmacological interventions. In contrast, surgical treatment, such as cataract extraction with intraocular lens implantation, has been well-established and proven to be highly effective and safe [8,9]. However, with increasing life expectancy and the uneven distribution of global development and medical resources, cataract not only diminishes patients' quality of life but also poses risks due to the rising demand for surgery, placing a significant economic burden on families and society [10]. Therefore, early diagnosis and treatment of cataract are effective strategies to reduce blindness caused by the condition. Furthermore, understanding current epidemiological patterns and forecasting future trends are crucial for developing healthcare policies aimed at mitigating such blindness.

At present, research on the epidemiology of cataract often focuses on national or regional scales. For instance, the Swedish National Cataract Register (NCR) contains data on over 2.4 million cataract surgeries performed between 1992 and 2021. During this period, the rate of cataract surgeries recorded in the NCR in Sweden increased from 3,700 cases to 12,800 cases [11]. Moreover, studies utilizing the Global Burden of Disease (GBD) database have examined the burden of cataract between 1990 and 2019. These findings reveal a significant increase in the burden of cataract over this period [12,13]. However, epidemiological studies on cataract that

focus on national or regional levels were limited in scope, particularly in terms of their coverage and sample size. Many of these studies were confined to specific regions or populations, which may not be representative of the broader global or national context. Additionally, the sample sizes of these studies were often small, leading to potential biases and limited generalizability of the findings. These limitations make it challenging to analyze the epidemiological trends of cataract from a broader perspective and to draw comprehensive conclusions about the disease burden at a global level. In contrast, while studies using the GBD 2019 data on cataract burden provide valuable insights, the GBD 2021 dataset, which includes epidemiological data from the Covid-19 era, offers a more current perspective on the global trends of cataract epidemiology.

Therefore, in this study, we utilized cataract burden data from GBD 2021 to estimate, analyze, and predict future trends of the disease burden globally, across GBD regions, and within countries. We examined variations in disease burden using parameters such as gender, age, and the Socio-Demographic Index (SDI), and employed the Bayesian Age-Period-Cohort (BAPC) model to project cataract prevalence and Disability Adjusted Life Years (DALYs) from 2022 to 2030. These detailed analyses aim to enhance understanding of the current and potential future burden of cataract, facilitating estimates of required medical resources and surgical needs, and thereby assisting governments and health authorities in developing more effective public health policies and resource allocation strategies.

## Materials and methods

### Data acquisition and disease classification

The Global Burden of Disease (GBD) study was a comprehensive research initiative aimed at systematically assessing the impact of major diseases and disabilities on global health. Compared to traditional epidemiological research methods, the GBD study offered advantages such as extensive coverage, systematic data organization, and standardized analysis. In the GBD study, the health burden of diseases could be quantified using measures such as Prevalence, Incidence, Deaths, Years Lived with Disability (YLDs), Years of Life Lost (YLLs), and Disability-Adjusted Life Years (DALYs).

In this study, data on the global burden of cataracts were obtained from GBD 2021 (https://ghdx.healthdata.org/gbd-2021). Additionally, this database is part of a global research project led by the Institute for Health Metrics and Evaluation (IHME) at the University of Washington, and its use has been approved by the university's review board. The study does not involve ethical issues related to patient information or informed consent, as previous studies have specifically clarified the ethical exemption of GBD [14].

Specifically, we selected the Prevalence and DALYs indicators for cataracts from 1990 to 2021 across 21 GBD regions and 204 countries, including data on both Number and Rate. We also selected cataract burden data categorized by Both, Female, and Male. Given that cataract has a low mortality rate and that in GBD 2021, DALYs were equivalent to YLDs, the following descriptions and analyses are based on DALYs. Moreover, the GBD 2021 database did not include mortality data for cataracts. Therefore, this study utilized Prevalence and DALYs as general indicators of cataract-related health impairment [12,15].

Furthermore, in the GBD 2021 study, cataracts were characterized as lens opacification caused by protein accumulation, which subsequently led to visual impairment (International Classification of Diseases, Tenth Revision (ICD-10) codes H25-H26.9, H28-H28.8) [16].

### Data analysis

Firstly, this study described the burden of cataracts in 21 GBD regions by comparing the absolute counts of cataract prevalence and DALYs from 1990 to 2021. Furthermore, it calculated the age-standardized rates (ASR) of prevalence and DALYs from 1990 to 2021 across these 21 GBD regions, accounting for the effects of population age structure to mitigate biases due to differences in age group proportions. Subsequently, the study assessed the Estimate Annual Percentage

Change (EAPC) in ASR from 1990 to 2021 to reflect the changes in ASR over this period (positive values indicating an increase and negative values a decrease). The prevalence and DALYs of cataracts in 204 countries for 2021 were then mapped on a global scale to visualize the burden of cataracts worldwide. Finally, the 95% confidence interval (CI), derived from the 25th and 975th values of 1,000 ordered samples, provided a range of reliability and precision for the estimates, indicating the potential variability of the results [12,14,17].Lastly, the R programming packages utilized in this analysis included dplyr, ggplot2, reshape2, and readxl.

Secondly, the SDI served as an indicator reflecting educational, healthcare, and economic conditions across different regions and countries, and it was extensively applied in global burden of disease and other epidemiological research. Specifically, SDI ranged from 0 to 1, with higher values indicating better socioeconomic development: high SDI countries (SDI > 0.81), upper-middle SDI countries (0.70 < SDI ≤ 0.81), middle SDI countries (0.60 < SDI ≤ 0.70), lower-middle SDI countries (0.46 < SDI ≤ 0.60), and low SDI countries (SDI ≤ 0.46) [14]. In this study, we categorized SDI according to the global SDI status provided by GBD into 21 GBD regions and 204 countries, and conducted correlation analyses on cataract prevalence, DALYs, and SDI. The scatter points for each region depicted changes from 1990 to 2021 from left to right [18]. In this section, the R programming packages utilized included reshape, ggplot2, and ggrepel.

Furthermore, we conducted a frontier analysis of DALYs across 204 countries or regions. Here, the frontier represented those countries or regions exhibiting the lowest burden of cataract disease, with SDI reflecting the minimal disease burden. The Effective Difference (EF) was defined as the gap between a region's disease burden and the frontier, indicating the difference between the disease burden shown by a country's or region's SDI and the potential achievable burden. Significant effective gaps suggested that there might be room for improvement in disease burden based on the country or region's position within the development spectrum. Consequently, this study utilized EF to assess the potential for reducing cataract DALYs across various regions [19].

Subsequently, the Bayesian Age-Period-Cohort (BAPC) model was employed as a statistical tool for predicting disease burden. This model integrated Bayesian statistical methods with an age-period-cohort analysis framework. It was developed from the traditional Age-Period-Cohort (APC) model. By accounting for the effects of age, period, and cohort on prevalence and DALYs, the BAPC model aimed to describe disease trends [20]. However, while the APC model was capable of forecasting future disease burden, its linear relationships among age, period, and cohort factors complicated parameter estimation. To address these complexities, the BAPC model utilized Bayesian techniques, combining prior information with sample data to estimate the posterior distribution of unknown parameters. This approach directly approximated the posterior distribution, resolving issues related to model mixing and convergence, thereby enhancing the model's reliability and accuracy [21,22]. In the BAPC model used in this study, we categorized individuals into age groups such as <9 years old, 20–24 years old, 25–29 years old, 30–34 years old, …, 75–79 years old, 80–84 years old, and >85 years old. We then projected the prevalence and DALYs of cataract disease for different genders up to the year 2030, focusing on both ASR and absolute counts [22]. Finally, the R programming packages utilized in this section included BAPC and INLA.

Finally, based on the major risk factors downloaded from the GBD 2021 website, this study selected metabolic risks, air pollution, and smoking, and estimated the proportion of the impact of these risk factors on cataract DALYs [23]. The R programming packages involved in this section were ggsci, dplyr, and ggplot2.

All statistical analyses conducted in this study were based on R (Version 4.2.3).

## Results

### Global burden of cataract DALYs and prevalence from 1990 to 2021

Globally, the absolute count of DALYs increased from 3.417 million in 1990 to 6.554 million in 2021, representing a 92.8% rise. Similar to the absolute count of prevalence, the increase in DALYs was more pronounced among females (Fig 1A). In contrast, the ASR of DALYs (ASDR) exhibited a declining trend, decreasing from 91 per 100,000 people in 1990 to

**Fig 1. Changes in cataract burden from 1990 to 2021.** A. The Number and Age-Standardized Rates of DALYs by gender; B. The Number and Age-Standardized Rates of Prevalence by gender.

77 per 100,000 people in 2021, with an EAPC of −0.4798 ([95% CI] −0.5766, −0.3828). Overall, although the absolute count increased, the ASDR exhibited a declining trend across all 21 GBD regions, with negative EAPC values. Notably, the regions with the most pronounced decline in ASDR were (EAPC: from −1.7691 to −1.1782): Andean Latin America, Southern Sub-Saharan Africa, Southeast Asia, Central Latin America, North Africa and the Middle East and South Asia (Table 1).

Furthermore, the absolute count of cataract prevalence increased from 42.332 million in 1990 to 100.571 million in 2021, marking an increase of nearly 138%. This prevalence exhibited a year-on-year increase, with a more pronounced rise among females (Fig 1B). Additionally, the ASR of prevalence (ASPR) rose from 1145 per 100,000 people in 1990 to 1181 per 100,000 people in 2021. The EAPC for prevalence was 0.2117 ([95% CI] 0.1172, 0.3063). Although the global burden of cataract prevalence increased, the ASPR declined in most regions compared to 1990. Regions with the most notable declines in ASPR included (EAPC: from −1.16 to −0.3578): Andean Latin America, Southeast Asia, and South Asia (Table 1).

Lastly, the results of the gender differences in the ASDR and ASPR across all age groups globally indicated significant gender disparities in South Asia, Oceania, Western Sub-Saharan Africa, Southeast Asia, and North Africa and the Middle East (S1 Fig). The age-standardized rates of both disease burden indicators were higher for females compared to males.

## Global burden of cataract across different SDI regions

As illustrated by Figs 2 and 3 reveal that the highest ASR for this disease occurred in regions with an SDI range of 0.3–0.5 (low-middle SDI). Notably, the regions and countries with pronounced disease burden in the low-middle SDI

**Table 1. Changes in the global and regional burden of cataract disease from 1990 to 2021.**

| | Prevalence | | | | | Disability-Adjusted Life Years | | | | |
|---|---|---|---|---|---|---|---|---|---|---|
| | 1990 counts (10k cases) | 2021 counts (10k cases) | 1990 Age standardized rate (per 100k population) | 2021 Age standardized rate (per 100k population) | Estimated Annual Percent Change (CI) | 1990 counts (10k cases) | 2021 counts (10k cases) | 1990 Age standardized rate (per 100k population) | 2021 Age standardized rate (per 100k population) | Estimated Annual Percent Change (CI) |
| Global | 4233.2 | 10057.1 | 1145 | 1181 | 0.2117 (0.1172-0.3063) | 341.7 | 655.4 | 91 | 77 | −0.4798 (−0.5766--0.3828) |
| Andean Latin America | 33.8 | 81.6 | 1814 | 1428 | −1.1600 (−1.312--1.0078) | 2.7 | 5.3 | 144 | 92 | −1.7691 (−1.8866--1.6515) |
| Australasia | 8.1 | 203 | 359 | 354 | 0.0039 (−0.0669-0.0748) | 0.5 | 1.1 | 22 | 20 | −0.2511 (−0.2894--0.2128) |
| Caribbean | 17.8 | 33.9 | 717 | 627 | −0.4336 (−0.4454--0.4217) | 1.5 | 2.3 | 59 | 43 | −1.0518 (−1.0726--1.031) |
| Central Asia | 52.8 | 82.5 | 1227 | 1147 | −0.2516 (−0.2851--0.2181) | 3.3 | 4.6 | 76 | 64 | −0.5969 (−0.6496--0.5442) |
| Central Europe | 56.7 | 91.7 | 408 | 394 | −0.1394 (−0.1564--0.1224) | 3 | 4.4 | 22 | 19 | −0.4284 (−0.4429--0.4138) |
| Central Latin America | 82.1 | 223.8 | 1115 | 935 | −0.5909 (−0.6097--0.5722) | 6.9 | 15.2 | 92 | 63 | −1.2913 (−1.3384--1.2441) |
| Central Sub-Saharan Africa | 7.6 | 18.3 | 456 | 444 | −0.0058 (−0.0327-0.021) | 0.6 | 1.2 | 34 | 27 | −0.6349 (−0.6874--0.5825) |
| East Asia | 578.5 | 2008.3 | 859 | 968 | 0.6428 (0.3926-0.8936) | 45.9 | 112.2 | 65 | 54 | −0.2433 (−0.5775-0.092) |
| Eastern Europe | 142.5 | 185.4 | 548 | 521 | −0.2248 (−0.2562--0.1934) | 9.1 | 10.3 | 35 | 29 | −0.6709 (−0.7291--0.6126) |
| Eastern Sub-Saharan Africa | 107.6 | 225.8 | 1608 | 1485 | −0.1815 (−0.2154--0.1476) | 10.4 | 19.5 | 148 | 123 | −0.5152 (−0.5545--0.4759) |
| High-income Asia Pacific | 54.1 | 153.8 | 292 | 282 | −0.056 (−0.0824--0.0296) | 3.7 | 8.8 | 19 | 17 | −0.3618 (−0.4000--0.3237) |
| High-income North America | 105.4 | 1939 | 291 | 284 | −0.0843 (−0.2053-0.0368) | 6.4 | 11.2 | 18 | 17 | −0.2339 (−0.3074--0.1604) |
| North Africa and Middle East | 251.9 | 628.7 | 1767 | 1571 | −0.4281 (−0.4578--0.3985) | 20.8 | 40.2 | 143 | 99 | −1.2804 (−1.3175--1.2433) |
| Oceania | 6.3 | 15.6 | 2650 | 2534 | −0.2083 (−0.3281--0.0883) | 0.5 | 1.1 | 194 | 161 | −0.6006 (−0.7065--0.4945) |
| South Asia | 1598.2 | 3674.7 | 3183 | 2661 | −0.3578 (−0.4893--0.2262) | 141.5 | 264 | 284 | 194 | −1.1782 (−1.2643--1.0921) |
| Southeast Asia | 550.3 | 1231.7 | 2502 | 2102 | −0.7234 (−0.7846--0.6621) | 42.9 | 77.8 | 192 | 132 | −1.3646 (−1.4532--1.276) |

*(Continued)*

**Table 1.** (Continued)

| | Prevalence | | | | | Disability-Adjusted Life Years | | | | |
|---|---|---|---|---|---|---|---|---|---|---|
| | 1990 counts (10k cases) | 2021 counts (10k cases) | 1990 Age standardized rate (per 100k population) | 2021 Age standardized rate (per 100k population) | Estimated Annual Percent Change (CI) | 1990 counts (10k cases) | 2021 counts (10k cases) | 1990 Age standardized rate (per 100k population) | 2021 Age standardized rate (per 100k population) | Estimated Annual Percent Change (CI) |
| Southern Latin America | 25.8 | 50.6 | 596 | 565 | −0.1392 (−0.1623--0.1161) | 1.7 | 2.7 | 38 | 30 | −0.6676 (−0.6883--0.6469) |
| Southern Sub-Saharan Africa | 27 | 46.3 | 1096 | 913 | −0.838 (−0.9102--0.7657) | 3.1 | 4.4 | 119 | 84 | −1.4866 (−1.5797--1.3933) |
| Tropical Latin America | 93.4 | 243.3 | 1181 | 980 | −0.1024 (−0.2741-0.0695) | 7.9 | 17.1 | 100 | 69 | −0.7823 (−0.9483--0.6161) |
| Western Europe | 265.5 | 453.8 | 453 | 433 | −0.1365 (−0.1494--0.1237) | 15.7 | 24.8 | 27 | 25 | −0.2873 (−0.3027--0.272) |
| Western Sub-Saharan Africa | 167.9 | 393.1 | 2136 | 2221 | −0.0026 (−0.1163-0.1113) | 13.6 | 27.2 | 171 | 151 | −0.4891 (−0.5945--0.3835) |

category included South Asia, Oceania, Southeast Asia, as well as Pakistan, Ethiopia, South Sudan, Afghanistan, and Cambodia. Among the 204 countries, South Asia exhibited the highest cataract disease burden, with its ASPR rising from approximately 200–300 per 100,000 people between 1990 and 2021, and its ASDR increasing from about 2500–3500 per 100,000 people over the same period. Additionally, Pakistan had the highest cataract disease burden among the 204 countries, with an ASPR of 308 cases per 100,000 people and an ASDR of 3448 cases per 100,000 people (detailed country-specific ASR can be found in S1 Table).

### Frontier analysis of cataracts

Frontier analysis revealed that as SDI increases, the ASDR decreases gradually (Fig 4). After an SDI greater than 0.5, changes in ASDR become relatively stable. The five countries/regions with the highest excess factor (EF range: 180.6–292.8) were Pakistan, Ethiopia, Nigeria, Cambodia, and Afghanistan. Conversely, the countries with the lowest EF (EF range: 10.3–21.7) were the Lithuania, Singapore, Denmark, the United Kingdom, and Iceland (detailed data are provided in S2 Table).

### Forecasting cataract disease burden from 2022 to 2030 based on the BAPC model

This study, based on the BAPC model, utilized the ASPR and ASDR prediction pie charts (Fig 5) to illustrate a gradual decline in cataract burden from 2022 to 2030 for both males and females, with a satisfactory fit of the pie chart model. Although the age-standardized cataract burden exhibited a decreasing trend, the absolute counts of cataract burden for both genders in 2030 were projected to increase compared to 2021. The prediction of age-specific numbers and rates based on the BAPC model shows the burden of cataract were predominantly concentrated in individuals aged 50 and above (S2 Fig). Therefore, interventions targeting middle-aged and older adults are essential to reduce disability caused by cataracts.

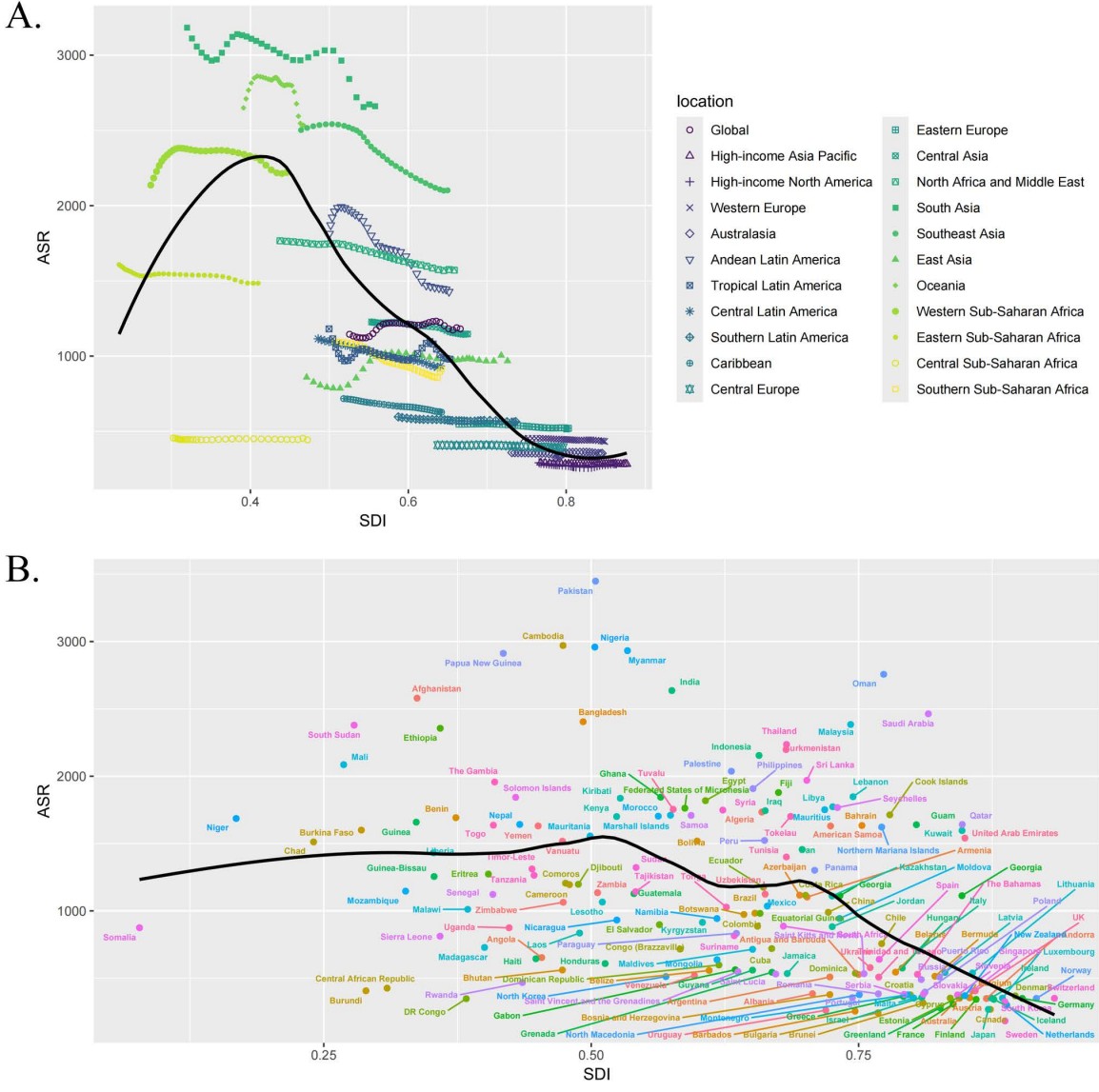

**Fig 2. The differences in ASPR of cataract among regions with different Socio-Demographic Index levels.** A. The ASPR of cataract in the 21 GBD regions; B. The ASPR of cataract in 204 countries.

## Analysis of risk factors influencing cataract DALYs

As illustrated in Fig 6, metabolic risks such as diabetes, obesity, and hormonal changes significantly impacted the DALYs of cataract patients, with metabolic risks accounting for 26.6% of the global cataract DALYs. Regions with particularly high proportions included North Africa and the Middle East (35.8%), High-income Asia Pacific (35.2%), and Central Latin America (33.7%). Additionally, air pollution, including PM2.5, heavy metal ions, and other forms of air pollution, also had a broad impact on cataract DALYs, affecting 29.8% of patients globally. The regions experiencing the most severe impact were Eastern Sub-Saharan Africa (52.2%), Oceania (47.4%), Western Sub-Saharan Africa (47.2%), and Central Sub-Saharan Africa (43.2%). Hence, we believe that reducing exposure to metabolic factors and air pollution is very likely to make a substantial contribution to reducing the burden of cataract.

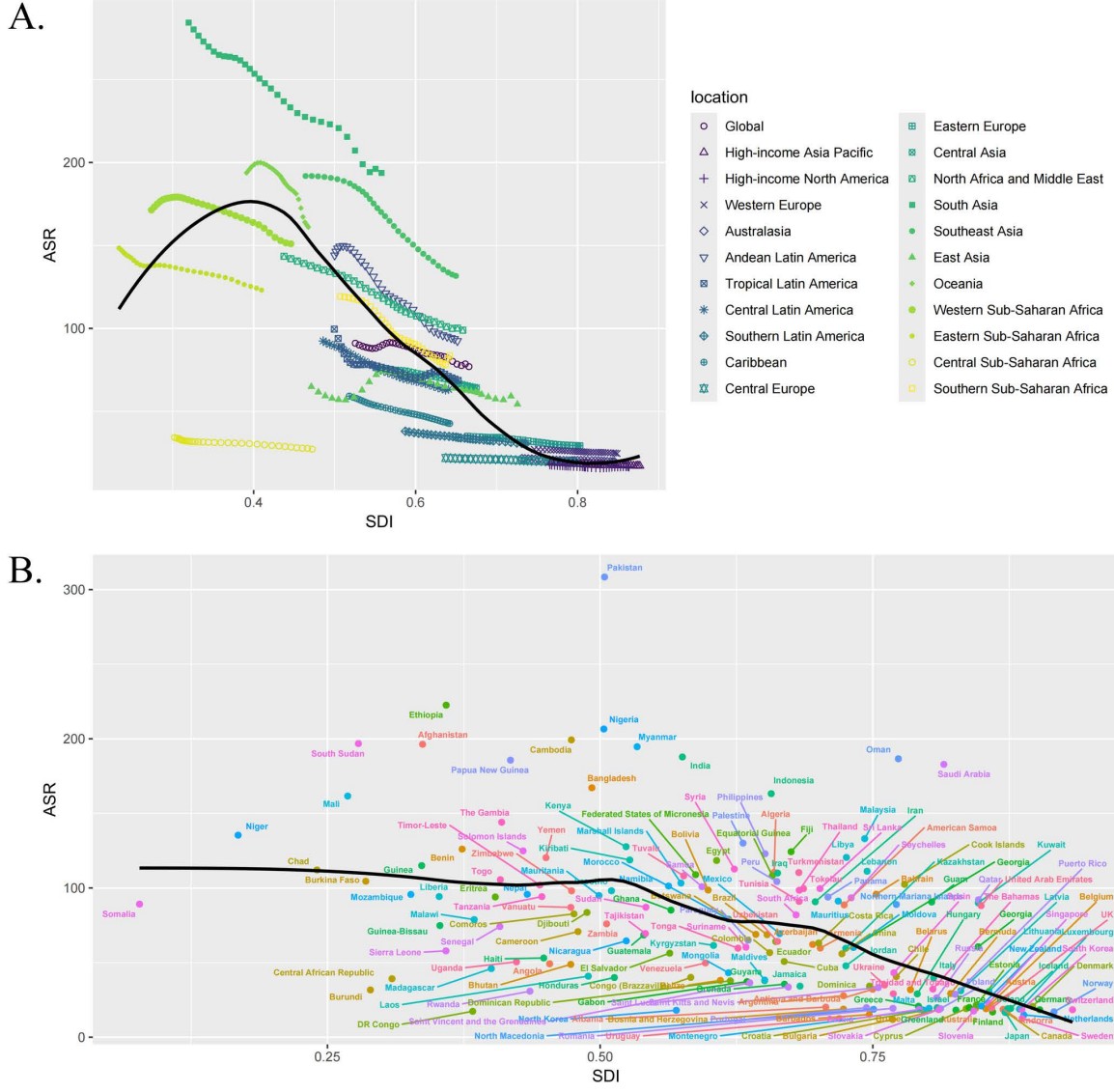

**Fig 3. The differences in ASDR of cataract among regions with different Socio-Demographic Index levels.** A. The ASDR of cataract in the 21 GBD regions; B. The ASDR of cataract in 204 countries.

## Discussion

This study utilized global burden of disease data on cataracts from the GBD 2021 database to analyze the changes in the disease burden of cataracts across various dimensions worldwide from 1990 to 2021. Our findings indicated that the ASDR due to cataracts exhibited a declining trend, while the ASPR showed an increasing trend. The higher burden of cataracts was concentrated in medium and low-middle SDI regions. Moreover, the advanced analysis demonstrated that there is considerable potential for improving DALYs in regions with lower SDI levels. Lastly, predictions based on the BAPC model also suggested a declining trend in the global burden of cataracts by 2030. The following sections will provide a detailed discussion of the results obtained from this study.

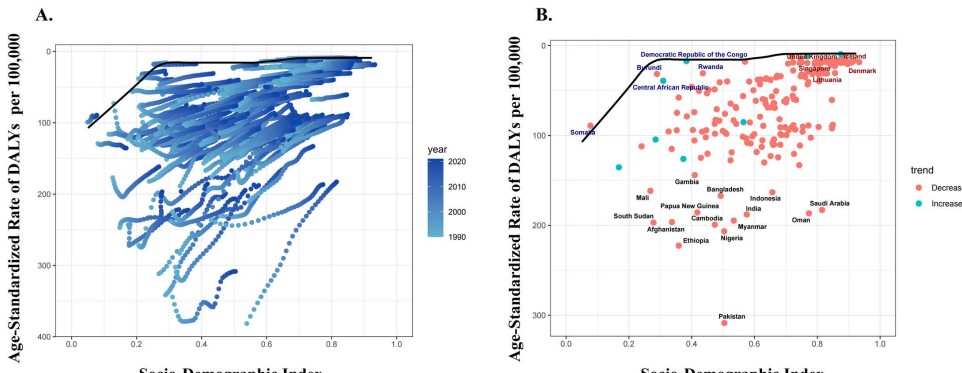

**Fig 4. Frontier analysis of Disability-Adjusted Life Years.** A. The distance of cataract ASDR from the "Frontier" in 204 countries during 1990-2021; B. The distance of ASDR from the "Frontier" in 204 countries in 2021.

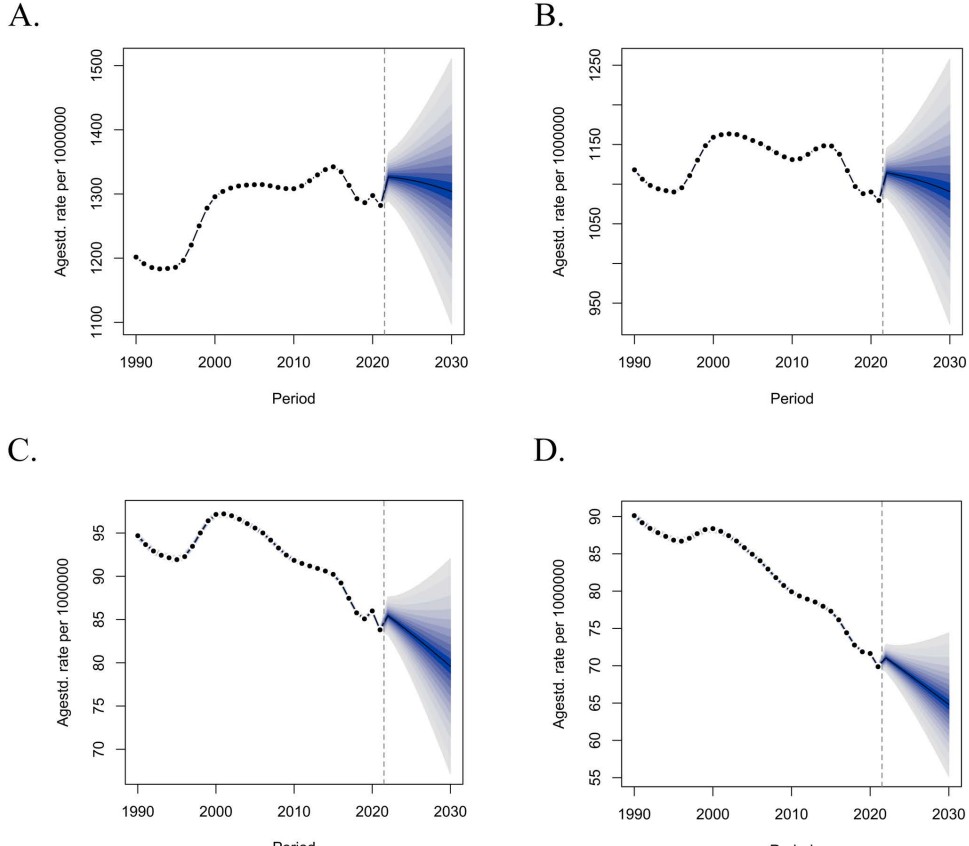

**Fig 5. Prediction of cataract disease burden from 2022 to 2030 based on a Bayesian-age-period-cohort-model.** A. Prediction of female prevalence; B. Prediction of female ASDR; C. Prediction of male prevalence; D. Prediction of male DALYs.

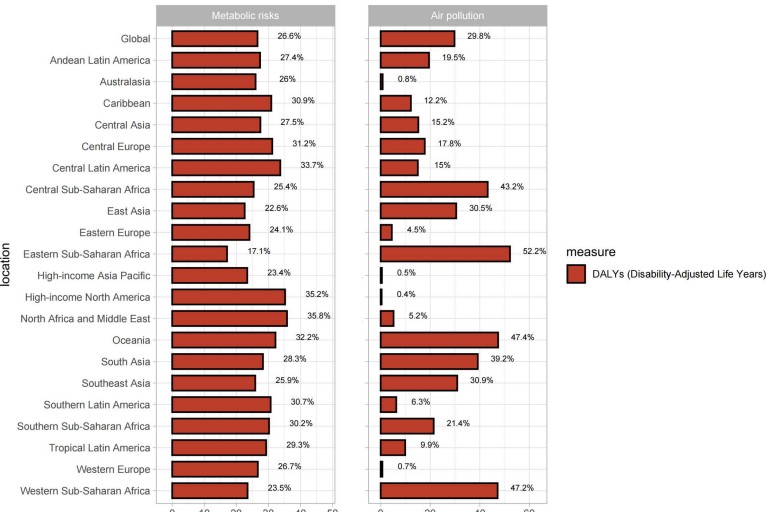

**Fig 6. Analysis of attributable risk factors for Disability-Adjusted Life Years.**

## Ages

Specifically, in our study, nearly 90% of cataract patients in our study were over the age of 40 (S2 Fig). At the same time, cataracts are widely recognized within the medical community as an age-related condition [24–27]. Cataracts not only pose a risk of blindness for elderly patients but also elevate the risk of falls and fractures within this population [28,29]. Research by Pundlik et al. has demonstrated that the fracture risk among cataract patients is significantly higher compared to other eye conditions, with increased risk observed across nearly all fracture sites [28,29]. Moreover, cataract-related visual impairment and blindness, particularly in the elderly, is associated with a heightened risk of psychological and mental health issues such as anxiety, depression, Alzheimer's disease, vascular dementia, and cognitive impairment, contributing to a high number of DALYs for these patients [30–33]. Therefore, allocating healthcare resources specifically for certain age groups is essential. This targeted approach can effectively reduce the incidence of blindness caused by cataracts.

## Genders

Our study also found significant gender differences in the burden of cataracts. Globally and across 21 GBD regions, the burden of cataracts is notably higher in females than males (S1 Fig). Although the ASR for cataract prevalence and DALYs showed a relatively stable and fluctuating trend approaching 2021, the absolute numbers are rising, predominantly among women. This gender inequality may be attributed to hormonal changes and societal roles. Researchers like Zetterberg have analyzed this imbalance, suggesting that estrogen might offer antioxidant protection against cataract formation, but a decline in estrogen levels post-menopause increases cataract risk in women [34]. Additionally, as individuals age, the fluctuations in female hormonal levels become more pronounced. Furthermore, clinical retrospective studies have identified that women of lower socioeconomic status and those with higher parity are at an increased risk of developing cataracts [24]. Moreover, the division of social responsibilities is also a significant factor. Relatively speaking, women are more frequently involved in cooking tasks in contemporary settings. Research by Rafiq et al. has demonstrated a positive correlation between indoor air pollution (resulting from the burning of biomass fuels) and cataracts in women, with the risk of disease increasing with age [27]. This disparity is particularly pronounced in middle and lower-middle SDI regions. For example, in Nigeria, the prevalence of cataracts among elderly women is higher compared to younger women and men, leading to a greater demand for cataract surgeries [35].

## Metabolism

The results of the risk factor analysis in this study reveal a significant association between cataract DALYs and metabolic-related issues. Factors including, but not limited to, diabetes, obesity, liver disease, and high-fat and high-sugar diet are associated with an increased risk of cataracts. Metabolic disorders often lead to higher cataract ASPR and ASDR.

Firstly, consider diabetes and obesity. Ye et al. analyzed data from the UK Biobank to investigate the relationship between the age of diabetes diagnosis and vision impairment. Their findings indicated that early diagnosis of type 2 diabetes (T2D) significantly increases the risk of cataracts and glaucoma, with individuals with type 1 diabetes (T1D) experiencing even higher risks [36]. Moreover, Yuan et al. applied Mendelian randomization to demonstrate that genetic traits related to high BMI, such as those associated with T2D, can raise the risk of cataracts [6]. Interestingly, Chen et al. used meta-analysis to show that moderate overweight (BMI: 25–29.9) may be negatively associated with cataract prevalence, whereas obesity (BMI: > 30) is positively associated with cataract prevalence [37]. Therefore, obesity and obesity-related diseases are now also considered significant risk factors for cataract development.

Furthermore, in the context of liver disease, Kang et al., based on a cross-sectional cohort study from the 2010–2011 Korean National Health and Nutrition Examination Survey, found that participants with cataracts had higher liver fibrosis scores. After data optimization, they observed that metabolic dysfunction-related fatty liver disease was significantly associated with an increased odds ratio for cataracts and suggested that fatty liver disease might act as an independent risk factor for cataract development [38].

Additionally, dietary habits have been shown to influence the risk of cataract development. Numerous studies suggest that a balanced diet can significantly reduce the risk of developing cataracts. For example, research by Kim et al. indicates that higher intake of carbohydrates, polyunsaturated n-6 fatty acids, vitamins, and minerals is associated with a lower risk of cataract incidence [39]. Similarly, Jiang et al. found and summarized that higher consumption of vegetables and fruits is negatively correlated with cataract prevalence, with this trend being more pronounced in older adults. Conversely, a high Dietary Inflammatory Index (DII) is positively associated with cataract Prevalence [40–42]. Specifically, Sesso et al. found that moderate intake of anthocyanins, vitamins B2, B12, and B6 is associated with a reduced risk of cataracts. Additionally, daily consumption of tea (4–6 cups) and coffee (2–3 cups) is linked to a lower risk of cataracts [43–45].

Thus, differences in dietary patterns and consumption abilities across regions with varying SDI levels may contribute to disparities in cataract disease burden. Specifically, middle and high SDI regions focus more on balanced nutrition, while low SDI regions may face either nutrient deficiencies or excesses, both of which increase cataract risk. Hence, dietary health significantly impacts cataract burden. We advocate for medical and national policymakers to address these nutritional issues in specific populations to reduce the burden of cataracts and other related diseases [46].

## Environmental factors

At the same time, environmental sanitation also plays a role in the burden of cataract disease. Ebrahimi et al. have described how elevated levels of heavy metals, such as lead (Pb), in the environment can directly or indirectly contribute to the formation of cataracts [47]. Moreover, the depletion of ozone in the atmosphere increases the levels of ultraviolet (UV) radiation. Several studies have indicated that increased UV exposure raises the risk of developing cataracts [48–50]. Similarly, air pollution caused by PM2.5 and NO2 is widely recognized in the field of ophthalmology as being positively correlated with the prevalence of cataracts [51,52]. And research by Alhasa et al. has found that radiation exposure in the workplace can also influence the risk of developing cataracts. For instance, healthcare workers who are exposed to radiation have a higher risk of developing posterior subcapsular cataracts [53].

Currently, factors such as global climate change, increased industrialization, and the transfer of polluting factories from developed to developing countries have led to various environmental pollutants in lower SDI regions. Although the ASR of disease burden are declining globally, the total number of cases continues to rise. Therefore, we advocate for the improvement of living environments, as it holds significant potential to reduce the burden of diseases, including cataracts.

## Coverage of cataract surgery

Currently, cataract treatment methods have reached a considerable level of maturity, with advanced approaches to surgical techniques, comprehensive patient management from preoperative assessment to postoperative care, and effective management of complications, access to high-quality healthcare services for cataract patients remains highly variable across the globe. In many low- and middle-income countries, the availability of effective cataract surgery is still limited due to factors such as a shortage of trained eye care professionals, financial barriers, and inadequate health infrastructure. This disparity highlights the significant challenges in ensuring equitable access to high-quality cataract care on a global scale [8,54].

Therefore, Cataract treatment through surgery has become quite advanced. However, a crucial factor is whether cataract patients can effectively access surgical treatment. Due to variations in the SDI, the quality of healthcare differs across regions. Research by McCormick et al. has revealed significant disparities in cataract surgical coverage between different SDI regions. For example, Hungary has the highest coverage rate at 70.3%, while Guinea-Bissau has the lowest at 3.8%. The median cataract surgical coverage rate in high-income countries is 60.5%, whereas it is only 14.8% in low-income countries [55]. In regions with a lower SDI, effective cataract surgical coverage faces numerous challenges, such as uneven distribution of healthcare resources. In countries like India, where SDI is low, the rate of effective cataract surgery coverage increases with higher education levels—from 31.0% among the illiterate to 59.7% for those who have completed 10 years of education [56]. In China, differences between urban and rural areas, as well as educational levels, contribute to disparities in cataract disease burden similar to those observed in different SDI regions [57]. Major reasons for the inability to access cataract surgery include cost, lack of perceived need, and fear [58]. Thus, it is crucial to update national blindness prevention strategies, enhance public awareness, and improve both the quality and coverage of cataract surgery.

Indeed, the data from 2020 and 2021, which includes cataract disease burden during the COVID-19 pandemic, highlights significant impacts on healthcare delivery. During the pandemic, widespread lockdowns and the reallocation of medical resources led to a notable decrease in cataract surgeries, as observed in regions such as Malaysia and South Africa. This reduction in surgical activity likely means that the cataract disease burden for these years could be underestimated [59,60]. The pandemic has emphasized the need for strategies to address the backlog and ensure continuity of care for cataract patients.

## Measures to reduce the burden of cataract disease

The "Vision 2020: The Right to Sight" initiative, launched in 1999 by the World Health Organization (WHO) in collaboration with the International Agency for the Prevention of Blindness (IAPB), aimed to eliminate avoidable blindness by 2020. Building on the progress made under Vision 2020, the IAPB has introduced the "2030 In Sight" strategy, which seeks to end avoidable blindness and ensure universal access to eye care by 2030. This new strategy is aligned with the WHO World Report on Vision and emphasizes the integration of eye health into broader health systems and the importance of addressing vision as a fundamental social and economic issue, aims to significantly reduce or even eradicate the global burden of blindness-inducing diseases [2,61]. Although Abdulhussein et al. have indicated that this initiative has had a positive impact on reducing the burden of cataract disease, challenges in addressing the cataract burden persist [62]. Therefore, it is crucial to undertake meaningful actions to mitigate the cataract disease burden. This includes implementing free Rapid Assessment for Avoidable Blindness (RAAB) screenings and other ocular disease assessments [63], aims to enable more cataract patients to receive surgical treatment earlier, thereby reducing the disease burden of cataracts. Additionally, leveraging advanced internet platforms to disseminate knowledge about cataract disease and enhance public awareness and understanding of cataracts is essential [64]. Promoting prenatal diagnostics and screening for high-risk genes associated with cataract formation can help reduce the disease burden. Additionally, it is important to address the polarization of EF values resulting from disparities in the SDI in frontier analysis results [65–67].

In conclusion, this study represents the first comprehensive analysis of cataract disease burden utilizing data from the GBD 2021 database, encompassing a global perspective, as well as detailed evaluations across 21 GBD regions and 204 countries. Additionally, the study employs the BAPC model to forecast the global cataract burden from 2022 to 2030. However, there are notable limitations to this research. Firstly, the study considers the overall cataract disease burden without differentiating between various types of cataracts. Secondly, due to the disparity in healthcare standards worldwide and variability in the availability of public health data, there is a possibility that the cataract burden may be underestimated.

## Conclusions

This study utilizes cataract disease burden data from 1990 to 2021 in the GBD 2021 database, analyzing global, regional, and national trends based on SDI, cutting-edge factors, risk determinants, and future burden predictions. Our findings indicate that while there may be a potential decline in the global cataract burden in the future, reducing this burden remains a significant challenge. Therefore, we urge global health policymakers to enhance public awareness about cataracts and promote early diagnosis and treatment to reduce the number of people affected by blindness worldwide.

## Supporting information

**S1 Fig.  Comparison of the cataract burden by gender globally and across 21 GBD regions.**
(TIF)

**S2 Fig.  Comparison of cataract burden by gender in 2021 and 2030 based on the BAPC model.**
(TIF)

**S1 Table.  The specific values of Age - Standardized Death Rate and Age - Standardized Prevalence Rate in 204 countries and regions.**
(XLSX)

**S2 Table.  The values of the "Frontier" and their distances under different Socio - Demographic Index levels.**
(XLSX)

## Acknowledgments

This study wants to thank the GBD 2021 collaborators.

## Author contributions

**Data curation:** Lixia Lin, Yongshun Liang, Peipei Liao.

**Formal analysis:** Lixia Lin, Yongshun Liang, Qingqiao Gan, Peipei Liao.

**Funding acquisition:** Hao Liang.

**Investigation:** Lixia Lin, Yongshun Liang, Qingqiao Gan, Hao Liang.

**Methodology:** Lixia Lin.

**Project administration:** Lixia Lin.

**Resources:** Guiyang Jiang, Tianqi Yang, Hao Liang.

**Software:** Guiyang Jiang, Tianqi Yang.

**Supervision:** Hao Liang.

**Writing – original draft:** Lixia Lin.

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
