## [Decision Letter · Decision Letter 0]

Dear Dr. Liang,

Thank you for submitting your manuscript to PLOS ONE. After careful consideration, we feel that it has merit but does not fully meet PLOS ONE’s publication criteria as it currently stands. Therefore, we invite you to submit a revised version of the manuscript that addresses the points raised during the review process.

**There are minor revisions suggested by the reviewers. **

We look forward to receiving your revised manuscript.

Kind regards,

Ejaz Ahmad Khan, M.D, MPH, FFPH

Academic Editor

PLOS ONE

**Journal Requirements:**

4. We note that Figure 2 in your submission contain map images which may be copyrighted. All PLOS content is published under the Creative Commons Attribution License (CC BY 4.0), which means that the manuscript, images, and Supporting Information files will be freely available online, and any third party is permitted to access, download, copy, distribute, and use these materials in any way, even commercially, with proper attribution. For these reasons, we cannot publish previously copyrighted maps or satellite images created using proprietary data, such as Google software (Google Maps, Street View, and Earth). For more information, see our copyright guidelines: http://journals.plos.org/plosone/s/licenses-and-copyright.

We require you to either present written permission from the copyright holder to publish these figures specifically under the CC BY 4.0 license, or remove the figures from your submission:

Reviewers' comments:

Reviewer's Responses to Questions

**Comments to the Author**

1. Is the manuscript technically sound, and do the data support the conclusions?

Reviewer #1: Yes

Reviewer #2: Partly

2. Has the statistical analysis been performed appropriately and rigorously?

Reviewer #1: Yes

Reviewer #2: Yes

3. Have the authors made all data underlying the findings in their manuscript fully available?

Reviewer #1: Yes

Reviewer #2: Yes

4. Is the manuscript presented in an intelligible fashion and written in standard English?

Reviewer #1: Yes

Reviewer #2: Yes

**Reviewer #1: ** Thank you for analyzing the global, national and regional trends of cataract.

Few of my comments are:

Please write full form of the words in abstract and before the start of sections.

Please align with PLOS one guideline, as the line number are missing in manuscript.

**Reviewer #2:**  REVIEW OF MANUSCRIPT: PONE-D-24-51666

1. Overall, the manuscript appears to be well-structured, thorough, and provides valuable insights into the “Global, regional, and national burden of cataract and projections from 1990 to 2021.Data provided by the researchers is adequate, however, it would be of more benefit if the discussion section provided a more comprehensive analysis of the implications of the research findings.

2. Abstract

The abstract provides a concise overview of the research conducted in this manuscript. However, the objective and the title seem not to agree . The result says that there is a global decline while conclusion says there is a global increase. The author needs to pay attention to the abstract to make it more coherent.

3. Introduction:

The introduction effectively tried to discuss the global trends on cataract and management of cataract.

a. Line 41-43: Statement is not entirely correct. Where is the place for age-related cataract as a major cause of visual impairment and blindness?

b. Line 45-46: True statement but what would be the reason for this? Readers need to know.

c. Line 46-49: Please include reference.

d. Line 60: Kindly clarify what you mean by scope and what the limitations were.

4. Methods:

The materials and methods section are comprehensive and well-structured however the information needs to be presented in a step wise manner for more clarity.

a. Line 85: Kindly change sex to gender.

b. Line 98-99: Check the acronym represented.

c. Line 106: Write SDI in full

d. Line 108-111: Include a reference

e. Line 126-128: Include a reference

f. Line 142-143: Rephrase “The increase in DALYs was primarily attributable to females”.

5. Result:

The results section is detailed having various subheadings that looked at several subheadings relevant to the paper.

a. Kindly number the tables/figures appropriately as figures and appendix are used interchangeably. Also, it is important not to include all the information on the studies included in the paper. Key /relevant finding should be included as prose, and this would significantly reduce the word count of the paper.

b. Line 227: Kindly rephrase

c. Line 229: “procedural pathways” Management of cataract starts from history taking to post op care. I would assume that procedural management means clinical guidelines. This doesn’t come out clear so you may ant to rephrase the line.

6. Discussion:

The discussion section of the manuscript provides a detailed overview of the study's findings regarding the prevalence and DALY as well as global trends in the burden of cataract. However, there are some aspects that could benefit from clarification, refinement, or expansion:

a. Line 229-230: Is this a global representation?

d. Line 231: Kindly expand on what situation is.

e. Line 234-235: This is a known fact and should be in your background.

f. Line 235-236: Kindly add reference.

g. Line 238-230: “Cataract-induced” Rather use cataract- related visual impairment and blindness or visual impairment and blindness from cataract particularly in the elderly”

h. Line 245: replace SEXES with GENDER

i. Line 250: replace Imbalance with “Gender inequality”

j. Line 266: What kind of dietary habits?

k. Line 278 & line 283: use a more scientific paper appropriate phrase to start the sentence.

l. Line 300: Environmental factors

m. Line 317:” Effective cataract surgical coverage” You should introduce this concept in the background and then analyse in results before the discussion.

n. Line 341-342: There is a new strategy called “2030 right to sight”. Vision 2020 is relevant however it is better to cite more recent publication “World Report on Vision” .

o. Line 346-347: RAAB is a survey, and it is related to Effective cataract surgical coverage

p. Line 350-351: Promoting prenatal diagnostics and screening for high-risk genes associated with cataract formation can help detect congenital cataracts however higher prevalence of cataracts are age related cataract.

7. References:

References are relevant to the topic and are recent papers,

Check the following references and ensure uniformity of referencing style that is journal appropriate.

5, 8, 15, 33.

**Do you want your identity to be public for this peer review?** For information about this choice, including consent withdrawal, please see our Privacy Policy

Reviewer #1: No

Reviewer #2: **Yes: ** Dilichukwu Isioma Aniemeka

---

## [Author Response · Author response to Decision Letter 1]

14 Feb 2025

Dear editors and reviewers,

We appreciate the valuable comments of the editor. Our point-by-point responses to these comments are as follows.

Notes from the editor:

When submitting your revision, we need you to address these additional requirements.Please ensure that your manuscript meets PLOS ONE's style requirements, including those for file naming. The PLOS ONE style templates can be found at https://journals.plos.org/plosone/s/file?id=wjVg/PLOSOne_formatting_sample_main_body.pdf and
https://journals.plos.org/plosone/s/file?id=ba62/PLOSOne_formatting_sample_title_authors_affiliations.pdf

Response: We sincerely appreciate the valuable comments and suggestions provided by the editorial board and reviewers regarding our manuscript. We have carefully reviewed the reviewer comments and thoroughly studied the PLOS ONE submission guidelines and formatting requirements.

In response to the editorial board's request concerning manuscript formatting, we have meticulously compared our manuscript against the provided style templates and have revised it accordingly. Specifically, we have:①Ensured that all files are named in accordance with PLOS ONE's guidelines.②Adjusted the title, author, and affiliation formatting based on the provided PLOS ONE template.③Modified the font, line spacing, paragraph formatting, and other stylistic elements in the main text to conform to the PLOS ONE template.④Carefully reviewed and corrected any other potential deviations from PLOS ONE's style requirements.

We believe that the revised manuscript now adheres to PLOS ONE's stylistic guidelines. Should any formatting issues remain, we would be grateful if you could bring them to our attention so that we may address them promptly. We thank the editorial board and reviewers again for their diligent work and insightful feedback.

We note that the grant information you provided in the‘Funding Information’ and‘Financial Disclosure’ sections do not match. When you resubmit, please ensure that you provide the correct grant numbers for the awards you received for your study in the ‘Funding Information’ section.

Response: We greatly appreciate the editorial board's careful review of our manuscript. We sincerely apologize for the discrepancy identified between the grant information provided in the 'Funding Information' and 'Financial Disclosure' sections, which resulted from an oversight on our part. We have now corrected the information in both sections to ensure accuracy. We again express our sincere apologies for any inconvenience this error may have caused. We are grateful for your prompt identification of this issue, which has helped us improve the quality of our manuscript.

Please include your full ethics statement in the ‘Methods’ section of your manuscript file. In your statement, please include the full name of the IRB or ethics committee who approved or waived your study, as well as whether or not you obtained informed written or verbal consent. If consent was waived for your study, please include this information in your statement as well.

Response: Thank you very much for your careful review of my manuscript. I have taken your suggestion to include a full ethical statement in the “Methods” section seriously and have revised it. According to your instructions, I have added the following to the “Methods” section of the manuscript (Line 98-102): “Additionally, this database is part of a global research project led by the Institute for Health Metrics and Evaluation (IHME) at the University of Washington, and its use has been approved by the university’s review board. The study does not involve ethical issues related to patient information or informed consent, as previous studies have specifically clarified the ethical exemption of GBD15. ” Thank you for your valuable suggestions, which will help improve the rigor and standardization of the manuscript.

We note that Figure 2 in your submission contain map images which may be copyrighted. All PLOS content is published under the Creative Commons Attribution License (CC BY 4.0), which means that the manuscript, images, and Supporting Information files will be freely available online, and any third party is permitted to access, download, copy, distribute, and use these materials in any way, even commercially, with proper attribution. For these reasons, we cannot publish previously copyrighted maps or satellite images created using proprietary data, such as Google software (Google Maps, Street View, and Earth). For more information, see our copyright guidelines: http://journals.plos.org/plosone/s/licenses-and-copyright.

Response: We sincerely appreciate the editorial board's thorough review of our manuscript. Regarding the potential copyright concerns associated with the map images in Figure 2, we fully understand the issue. In accordance with PLOS ONE's copyright policy, and to avoid any potential copyright infringement while ensuring smooth publication, we have decided to remove Figure 2 from the revised manuscript. While the removal of Figure 2 may have some impact on the presentation of certain results, we believe this is the most prudent approach to ensure compliance with PLOS ONE's publication requirements. We understand and respect PLOS ONE's stringent copyright policy, and we thank you for promptly identifying this potential risk

Please include captions for your Supporting Information files at the end of your manuscript, and update any in-text citations to match accordingly. Please see our Supporting Information guidelines for more information: http://journals.plos.org/plosone/s/supporting-information.

Response: We are grateful to the editorial team for their meticulous review of our manuscript and their valuable feedback. Regarding the suggestion to include captions for Supporting Information files and update in-text citations accordingly, we have addressed this point thoroughly. Detailed captions for all Supporting Information files have been added to the end of the manuscript(Line 647-654). Furthermore, we have carefully reviewed the manuscript to ensure complete consistency between in-text citations and the newly added Supporting Information captions. We believe the revised manuscript now fully meets PLOS ONE's requirements for Supporting Information. Thank you for this important suggestion, which significantly enhances the clarity and readability of our manuscript.

Response: We deeply appreciate the reviewers' meticulous evaluation and insightful comments on our manuscript. We apologize for any inconsistencies in the references and have meticulously revised the reference list to adhere to standard formatting guidelines. We sincerely thank the reviewers for their suggestions and will diligently revise the manuscript to enhance its overall quality. We are confident that these revisions will result in a more rigorous and standardized submission.

Reviewer #1: 

Thank you for analyzing the global, national and regional trends of cataract.

Few of my comments are:

Please write full form of the words in abstract and before the start of sections.

Please align with PLOS one guideline, as the line number are missing in manuscript.

Response: We are deeply grateful for your thorough review and insightful comments on our manuscript. We sincerely appreciate the time and effort you dedicated to analyzing our work, and your positive assessment is highly encouraging.

We have carefully considered your specific suggestions and will promptly address them:

Abbreviations: We have meticulously reviewed the abstract and the beginning of each section to ensure that all abbreviations are defined upon their first appearance. We will also conduct a comprehensive review of the entire manuscript to prevent any omissions.

PLOS ONE Guidelines: We sincerely apologize for the absence of line numbers. We will immediately reformat the manuscript to fully comply with the PLOS ONE submission guidelines, ensuring correct line number display on all pages.

We are extremely grateful for your valuable suggestions, which will significantly improve the quality of our manuscript. We will complete the revisions as soon as possible and resubmit the revised version. We are hopeful that the revised manuscript will meet the publication requirements. Thank you once again for your diligent work.

Reviewer #2: REVIEW OF MANUSCRIPT: PONE-D-24-51666

1.Overall, the manuscript appears to be well-structured, thorough, and provides valuable insights into the “Global, regional, and national burden of cataract and projections from 1990 to 2021.” Data provided by the researchers is adequate, however, it would be of more benefit if the discussion section provided a more comprehensive analysis of the implications of the research findings.

Response: We are most grateful for your thorough review and valuable feedback on our manuscript. Your positive assessment of the manuscript's structure, comprehensiveness, and insightful contributions is highly appreciated.

We fully concur with your recommendation to enhance the discussion section by providing a more comprehensive analysis of our findings. We acknowledge the importance of the discussion in interpreting the significance and value of our research, and we recognize that our previous manuscript lacked sufficient depth in this area.

In the revised manuscript, we have provided a more in-depth analysis of the research findings, exploring the reasons for variations in cataract burden across different regions and countries, as well as the underlying socioeconomic, demographic, and environmental factors contributing to these disparities.We have further examined the implications of our findings for public health policy, such as how to develop more targeted cataract prevention strategies, optimize the allocation of healthcare resources, and increase public awareness and understanding of cataract prevention.

We are confident that the revised discussion section will be more comprehensive and insightful, better elucidating the significance and value of our research results. Thank you once again for your invaluable feedback.

2. Abstract

The abstract provides a concise overview of the research conducted in this manuscript. However, the objective and the title seem not to agree . The result says that there is a global decline while conclusion says there is a global increase. The author needs to pay attention to the abstract to make it more coherent.

Response: We are deeply grateful for your meticulous review and valuable feedback on our manuscript. We sincerely apologize for the logical inconsistencies identified in the abstract and will promptly revise it to ensure coherence and accuracy. We are re-examining our research findings to guarantee that the results presented in the abstract align precisely with the data and analyses presented in the main text(Line 29-38). We will avoid any ambiguous or potentially misleading language. We are confident that the revised abstract will be more coherent and accurate, effectively engaging readers with our research. Thank you again for your invaluable insights.

3. Introduction:

The introduction effectively tried to discuss the global trends on cataract and management of cataract.

a. Line 41-43: Statement is not entirely correct. Where is the place for age-related cataract as a major cause of visual impairment and blindness?

b. Line 45-46: True statement but what would be the reason for this? Readers need to know.

c. Line 46-49: Please include reference.

d. Line 60: Kindly clarify what you mean by scope and what the limitations were.

Response: We are deeply grateful for your thorough review and invaluable feedback on our manuscript. We are particularly honored by your positive assessment of the introduction and sincerely appreciate the specific concerns you raised, which have been instrumental in improving the manuscript's quality.

We have carefully addressed the issues you identified in the introduction and will outline the revisions below:

a. Line 46-47: We thank you for pointing out the omission regarding the importance of age-related cataract as a leading cause of visual impairment and blindness when describing global cataract trends. We have included a discussion of age-related cataract in the revised manuscript and modified the phrasing to ensure a more accurate and comprehensive representation of this critical factor.

b. Line 51-56: We fully agree with your suggestion to explain the reasons behind the "true statement." To provide readers with a better understanding of the study's context, we have added explanations regarding the potential underlying causes of this phenomenon in the revised manuscript.

c. Line 59: We greatly appreciate you identifying the lack of references. We have added relevant references to support the statements made in this section.

d. Line 70-76: Following your recommendation, we have provided a clearer delineation of the study's scope and limitations in the revised manuscript.

We are confident that these revisions effectively address your concerns and significantly enhance the quality of the manuscript. Thank you once again for your diligent work and insightful suggestions!

4. Methods:

The materials and methods section are comprehensive and well-structured however the information needs to be presented in a step wise manner for more clarity.

a. Line 85: Kindly change sex to gender.

b. Line 98-99: Check the acronym represented.

c. Line 106: Write SDI in full

d. Line 108-111: Include a reference

e. Line 126-128: Include a reference

f. Line 142-143: Rephrase “The increase in DALYs was primarily attributable to females”.

Response: We are most grateful for your meticulous review and valuable feedback on the methods section of our manuscript. Your positive assessment of its comprehensiveness and clear structure is highly encouraging. We have carefully considered your specific suggestions and will implement the necessary revisions in the revised manuscript. To enhance clarity, we will strive to organize the methods section in a more structured, step-wise manner.

Line 81 and 157: We have replaced "sex" with "gender" to ensure accurate and appropriate terminology. Thank you for pointing out this nuance.

Line 118-119: We have thoroughly checked the abbreviations in this section to ensure their clarity and accuracy. We have corrected any instances where errors or ambiguities existed.

Line 82: We have expanded the abbreviation SDI to its full form, "Sociodemographic Index," when it first appears in the introduction section.

Line 109, 112 and 124: We have added relevant references to this section to support our statements.

Line 170: We have rephrased "the increase in DALYs was more pronounced among females" to more accurately reflect the study findings and avoid ambiguity.

We are confident that the revised manuscript will be more rigorous, standardized, and better suited for publication in journal. Thank you again for your diligent work and invaluable recommendations!

5. Result:

The results section is detailed having various subheadings that looked at several subheadings relevant to the paper.

a. Kindly number the tables/figures appropriately as figures and appendix are used interchangeably. Also, it is important not to include all the information on the studies included in the paper. Key /relevant finding should be included as prose, and this would significantly reduce the word count of the paper.

b. Line 227: Kindly rephrase

c. Line 229: “procedural pathways” Management of cataract starts from history taking to post op care. I would assume that procedural management means clinical guidelines. This doesn’t come out clear so you may ant to rephrase the line.

Response: We are most grateful for you

---

## [Editor Report · Decision Letter 1]

Global, regional, and national burden of cataract: A comprehensive analysis and projections from 1990 to 2021

PONE-D-24-51666R1

Dear Dr. Liang,

We’re pleased to inform you that your manuscript has been judged scientifically suitable for publication and will be formally accepted for publication once it meets all outstanding technical requirements.

Kind regards,

Osamudiamen Cyril Obasuyi, MD, MSc, FWACS, FMCOPh

Academic Editor

PLOS ONE

---

## [Editor Report · Acceptance letter]

PONE-D-24-51666R1

PLOS ONE

Dear Dr. Liang,

I'm pleased to inform you that your manuscript has been deemed suitable for publication in PLOS ONE. Congratulations! Your manuscript is now being handed over to our production team.

Kind regards,

on behalf of

Dr. Osamudiamen Cyril Obasuyi

Academic Editor

PLOS ONE